# Nucleosome dynamics render heterochromatin accessible in living human cells

Hemant K. Prajapati [1,2] ✉, Zhuwei Xu [1,2], Peter R. Eriksson[1,2] &
David J. Clark [1] ✉

The eukaryotic genome is packaged into chromatin, which is composed of a nucleosomal filament that coils up to form more compact structures. Chromatin exists in two main forms: euchromatin, which is relatively decondensed and enriched in transcriptionally active genes, and heterochromatin, which is condensed and transcriptionally repressed. It is widely accepted that chromatin architecture modulates DNA accessibility, restricting the access of sequence-specific, gene-regulatory, transcription factors to the genome. However, the evidence for this model derives primarily from experiments with isolated nuclei, in which chromatin remodeling has ceased, resulting in a static chromatin structure. Here, using a DNA methyltransferase to measure accessibility in vivo, we show that both euchromatin and heterochromatin are fully accessible in living human cells, whereas centromeric α-satellite chromatin is partly inaccessible. We conclude that all nucleosomes in euchromatin and heterochromatin are highly dynamic in living cells, except for nucleosomes in centromeric chromatin.

Chromatin consists of repeating units called nucleosomes, which contain ~147 bp of DNA coiled around a central histone octamer core. The octamer is composed of two molecules each of H2A, H2B, H3 and H4[1]. Nucleosomes are regularly spaced on the DNA, resembling beads on a string. This nucleosomal filament undergoes additional compaction to form euchromatin or heterochromatin. In mammalian cells, heterochromatin may be constitutive, involving the same genomic regions in all cells (typically gene-poor regions composed of short repeated sequences), such as pericentromeric and telomeric regions, or facultative, involving cell type-specific fully repressed genes[2–12]. Heterochromatin is associated with specific post-translational histone modifications: constitutive heterochromatin is marked by H3K9me2/3, whereas facultative heterochromatin is marked by H3K27me3, although there is some overlap[13]. Conversely, other histone marks, such as H3K4me1, H3K4me3, H3K36me3 and H3K27ac, are generally associated with euchromatin.

The highly condensed nature of heterochromatin suggests that access to the DNA may be limited or even prevented. However, large proteins and dextrans can penetrate heterochromatin domains to some extent when injected into living cells, suggesting that heterochromatin may be accessible[14]. Furthermore, heterochromatin protein 1 (HP1), which binds to H3K9me3 in constitutive heterochromatin, is mobile in living mammalian cells[15,16] and transcription of repeat sequences in constitutive heterochromatin occurs at low levels[7,17]. These data indicate that constitutive heterochromatin is at least partially accessible some of the time. Liquid-liquid phase separation may also be important in constitutive heterochromatin, resulting in exclusion of specific proteins from the heterochromatin phase[10,12,18–21]. These studies have led to a more nuanced view concerning the accessibility of constitutive heterochromatin.

Facultative heterochromatin contains inactive genes that are subject to Polycomb-mediated repression and are marked by

[1]Division of Developmental Biology, Eunice Kennedy-Shriver National Institute of Child Health and Human Development, National Institutes of Health, Bethesda, MD 20892, USA. [2]These authors contributed equally: Hemant K. Prajapati, Zhuwei Xu, Peter R. Eriksson. ✉e-mail: hemant.prajapati@nih.gov; clarkda@mail.nih.gov

H3K27me3 (reviewed by ref. 11). Genome-wide MNase-seq and ATAC-seq studies on isolated nuclei from various organisms have shown that inactive genes lack nucleosome-depleted regions (NDRs) at their promoters, unlike active genes. This observation suggests that nucleosomes prevent transcription factor binding at regulatory elements, such as promoters and enhancers, resulting in repression[22–26]. However, inactive promoters are partially accessible in mouse liver cell nuclei[27]. Although most transcription factors cannot access their cognate binding sites when incorporated into a nucleosome[22], there is a class of transcription factor, the "pioneer" factors, which bind to a nucleosomal site with high affinity[28]. Pioneer factors may be critical for initiating the process of nucleosome removal from regulatory elements by facilitating the binding of other transcription factors and recruitment of ATP-dependent chromatin remodelers to remove or displace blocking nucleosomes[29–31].

These observations suggest that nucleosomes play a crucial role in gene regulation by controlling access to regulatory elements. However, they are based primarily on experiments with nuclei, which may not be representative of chromatin in living cells. Indeed, we have shown recently that the budding yeast genome is globally accessible in living cells, except for the point centromeres and the silenced loci[32]. However, budding yeast chromatin is virtually all euchromatin, and lacks heterochromatin resembling that found in higher eukaryotes.

Here, we have asked whether human euchromatin is generally accessible in living cells, like that of yeast, and whether human heterochromatin is inaccessible, as might be expected. We measure genome accessibility at all GATC sites in living human MCF7 and MCF10A cells, using an adenovirus vector to express the sequence-specific *dam* DNA adenine methyltransferase. We find that the human genome is globally accessible in living cells, unlike in isolated nuclei. Remarkably, both constitutive and facultative heterochromatic sites are methylated only marginally more slowly than euchromatic sites. In contrast, sites in centromeric chromatin are methylated slowly and are partly inaccessible. Our data indicate that all nucleosomes in euchromatin and heterochromatin are highly dynamic in living cells, whereas nucleosomes in centromeric α-satellite chromatin are static. A dynamic architecture implies that simple occlusion of transcription factor binding sites by chromatin is unlikely to be critical for gene regulation.

## Results

### Global accessibility in live human cells
We adapted our qDA-seq method to measure genome accessibility in human cells[27,32]. Specifically, we used *E. coli dam* methyltransferase (Dam) as a probe for the accessibility of GATC sites in chromatin. Dam methylates the 'A' in GATC to '6mA'. It is challenging to express Dam without leaky expression using inducible promoter systems[33]. Therefore, we employed an adenovirus vector to transduce Dam fused to GFP and three HA tags into MCF7 cells (a human breast cancer cell line). Following transduction, cells were collected at various time points to monitor the kinetics of genome methylation (Fig. 1a) and of Dam production (Fig. 1b). Genomic DNA was purified and digested with DpnI, a restriction enzyme that cuts at GATC only if the 'A' is methylated on both strands. Agarose gel analysis of the extent of DpnI digestion revealed that the genome became almost fully methylated over time (Fig. 1c). Thus, Dam can access a large fraction of the human genome.

The conclusion that most of the genome is accessible is further supported by genomic analysis of GATC sites (Fig. 1d). The DpnI-digested DNA was sonicated to small fragments and subjected to paired-end sequencing. For each GATC site, the fraction methylated is calculated as the number of left or right DNA fragment ends divided by the coverage of that GATC site (Fig. 1a; see Methods). We calculated the methylation kinetics for each of the 5.8 million GATC sites in the human genome. To visualise the data for all GATC sites, we plotted the

fraction methylated for the median GATC site, and for all sites within the 5–95% methylated range, as a function of time after transduction (Fig. 1d). The median GATC site was ~80% methylated after 72 h and still trending upwards; 90% of all GATC sites show a similar trend.

We compared the methylation kinetics of transcriptionally active and inactive genes. Analysis of published ATAC-seq data for MCF7 cells[34] (Supplementary Fig. 1a) delineated two gene classes: one with high ATAC signal at the promoter, indicating the presence of an NDR, and one with low or no ATAC signal, indicating the absence of an NDR (Supplementary Fig. 1a). To confirm this interpretation, we performed MNase-seq on MCF7 nuclei and sorted the genes according to ATAC signal (Supplementary Fig. 1a). We observed a clear correlation between ATAC signal and the presence of a promoter NDR. Published gene expression (RNA-seq) data for MCF7 cells[34] also correlate with ATAC signal (Supplementary Fig. 1a).

We examined GATC site methylation at active and inactive promoters by plotting the mean GATC site methylation as a function of distance from the transcription start site (TSS). After 12 h of transduction, active genes show a weak nucleosome phasing signal that is exactly out of phase with our nucleosome dyad data (MNase-seq) for MCF7 nuclei (grey profile) (Fig. 1e; Supplementary Fig. 1b). This suggests that Dam methylation of the linkers and in promoter NDRs is slightly faster than methylation within the first (+1) and second (+2) nucleosomes. Mean methylation increases with time, reaching ~90% by 72 h. Promoter NDRs are methylated faster than gene bodies (Fig. 1e). Inactive genes show the same trend, but are methylated slightly more slowly, reaching ~83% after 72 h (Fig. 1f; Supplementary Fig. 1b). The methylation rate is almost uniform across inactive promoter regions, with no phasing; promoters and gene bodies are methylated at almost the same rate, consistent with the MNase-seq data (Fig. 1f). Nucleosome positioning appears to be essentially random around inactive promoters. We conclude that both active and inactive promoter regions are almost entirely accessible to Dam in vivo.

To determine whether higher transcription renders genes more accessible, we divided the active genes into quintiles according to their mRNA levels in MCF7 cells[34]. Quintile 5 has the most active genes; inactive genes were placed in a separate group. We observed a small but reproducible trend of increasing median methylation with increasing transcriptional activity; the most active genes were methylated marginally faster than the least active genes (Fig. 1g; Supplementary Fig. 1c). Thus, higher transcription correlates with a modest increase in methylation rate. However, the overarching conclusion is that both active and inactive genes are accessible to Dam.

### Slow, limited methylation at centromeres
To measure the accessibility of other genomic regions, we compared median methylation rates for GATC sites in promoters, gene bodies, tRNA genes, enhancers, CpG islands, silencers, replication origins and centromeres (Fig. 1h) using the hg38 genome annotations[35]. All regions are methylated at similar rates and to high levels, similar to gene bodies, except for the tRNA genes, which are methylated even faster, and the centromeres, which are methylated much more slowly and appear to be reaching a limit (Fig. 1h; Supplementary Fig. 1d,e). We quantified the median methylation rates for the various regions relative to the median for all genomic GATC sites by plotting the log of the unmethylated fraction as a function of time after transduction (Supplementary Fig. 1f). Relative methylation rates for promoters, enhancers, silencers and CpG islands were all slightly faster (1.2 to 1.7x the genomic median; see replicates) than gene bodies and replication origins (1.0 to 1.1x). tRNA genes were methylated ~2x faster, whereas centromeres were methylated ~2.5x more slowly (0.4x the genomic median). Thus, the range in median methylation rate is ~6-fold, from centromeres (slowest) to tRNA genes (fastest). We conclude that all genomic regions examined are fully accessible, except for the centromeres.

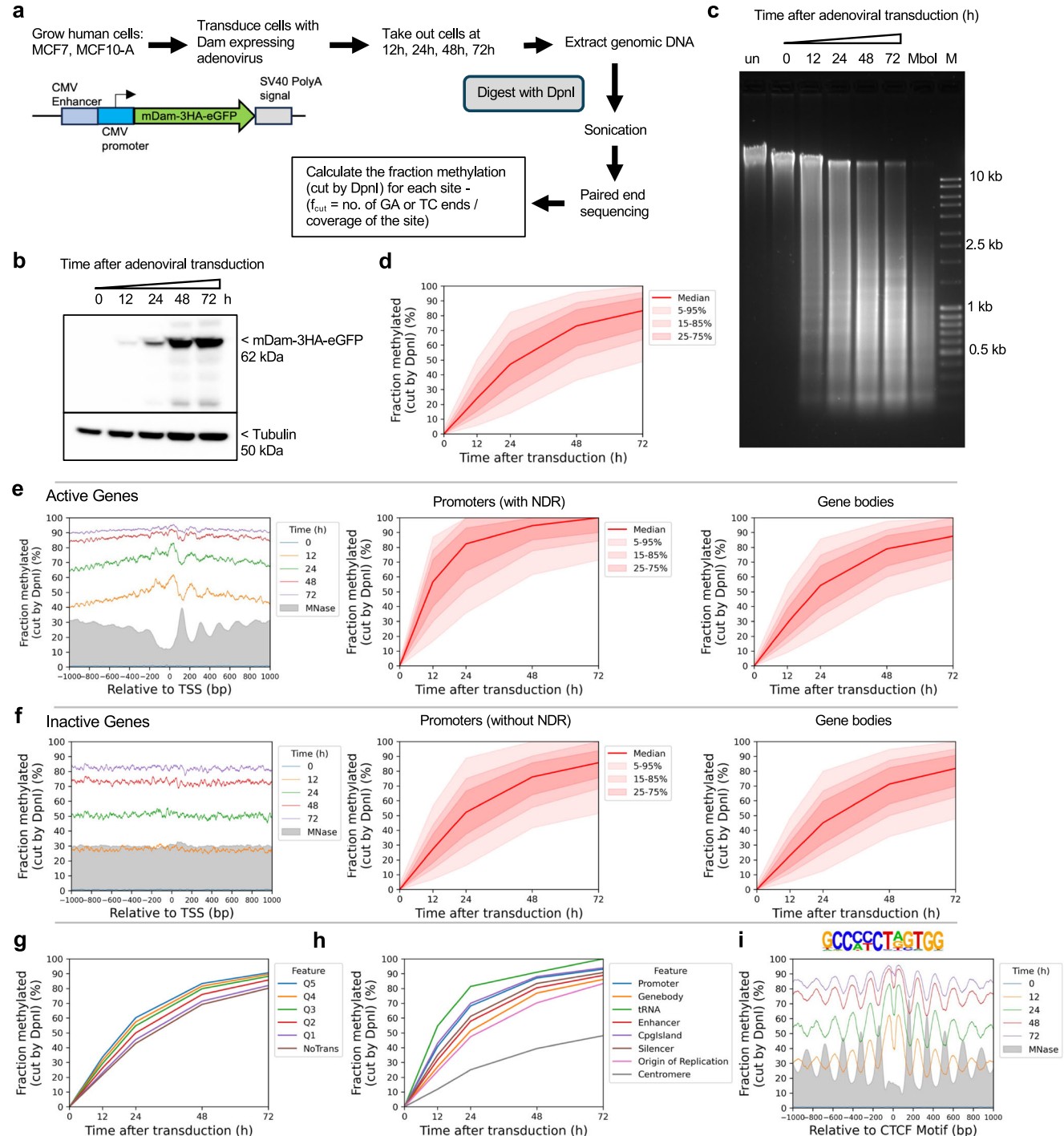

**Fig. 1 | The human genome is globally accessible in living MCF7 cells.**
**a** Schematic of adenovirus transduction and time course experiment to express Dam methylase in live cells. **b** Anti-HA immunoblot to detect Dam-3HA-eGFP expression in MCF7 cells. **c** Agarose gel electrophoresis of DpnI-digested genomic DNA purified from MCF7 cells as a function of time of adenovirus treatment. 'un', undigested genomic DNA; 'MboI', DNA from non-transduced cells digested with MboI as a marker for complete digestion at GATC sites; M, DNA size marker. **d** Almost complete methylation of GATC sites in MCF7 cells after transduction. Red line and shading: median GATC site methylation with data range indicated. **e, f**

Nucleosome phasing with respect to the TSS for active and inactive genes, as defined by ATAC-seq[34]. Grey profile: nucleosome dyad distribution in nuclei (MNase-seq data for MCF7 cells arbitrarily normalised to 30%). **g** The effect of transcriptional activity on methylation rate. Active genes were divided into quintiles Q1 to Q5 based on increasing transcriptional activity (Q5 is the highest) using RNA-seq data from ref. [34]; methylation of the median GATC site in each quintile is shown. Inactive genes are treated as a single separate group ('NoTrans'). **h** Median GATC methylation for various genomic regions. **i** Nucleosome phasing around CTCF motifs using the motif shown.

To search for large regions of relatively inaccessible chromatin at the chromosome level, we constructed a heat map showing the mean methylation rate for each 100 kb window along each chromosome. The T2T (Telomere to Telomere) human genome was used for this analysis because the centromeric regions have been thoroughly annotated[36]. The rate was calculated as above, using the mean methylation for all GATC sites in each window for each time point. We found that windows of slow methylation tend to cluster predominantly

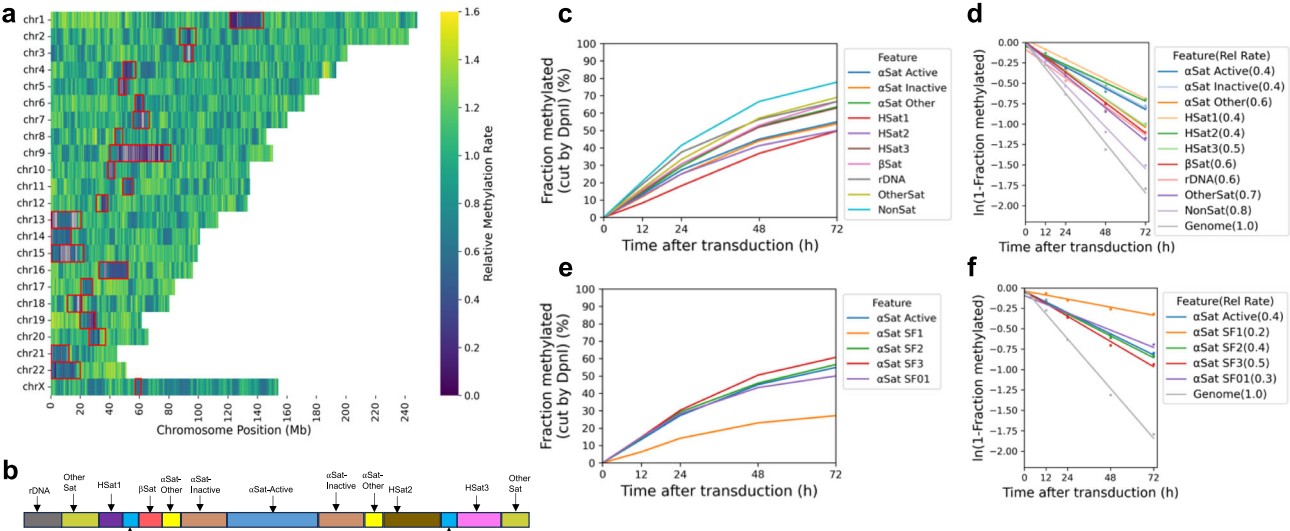

**Fig. 2 | Centromeres are methylated slower and reach a limit, unlike other genomic regions. a** Heat map showing the variation in methylation rate at the chromosomal level. The average methylation rate was calculated for all GATC sites in each 100 kb window in the T2T genome by plotting 'ln (1- fraction methylated)' against time after adenovirus transduction, and then normalised to the genomic average rate to obtain relative rates. Red rectangles: centromeric regions. **b** Schematic of the organisation of the various centromeric elements defined in the T2T genome assembly (adapted from ref. 37). Divergent α-Sat and monomeric α-Sat were combined as "α-Sat other". **c** Median GATC methylation time courses and **d**, methylation rates for the various centromeric elements. **e**, Median GATC methylation time courses and **f**, methylation rates for the various active supra-chromosomal α-satellite families.

at a few specific regions in each chromosome (Fig. 2a). Most of these windows are situated within centromeres (Fig. 2a; red rectangles indicate centromeres). The T2T genome has revealed the complexity of human centromeres, which make up ~6% of the genome and include many different repeats, including α-satellite repeats, human satellites (HSat1 to 5), β-satellite (βSat), γ-satellite (γSat), as well as non-satellite DNA[37] (Fig. 2b). The α-satellite repeats comprise variants of a ~171 bp repeat, which can be divided into 20 supra-chromosomal families. Active α-satellite repeats are enriched in centromeric histone H3 (CENP-A) and associate with the kinetochore.

All of these centromeric regions are methylated more slowly (0.4x to 0.7x) than the genomic average (1.0) (Fig. 2c, d); α-satellite and HSats exhibit the slowest methylation rates (Fig. 2d; Supplementary Fig. 2). We examined the methylation rates within the CENP-A-enriched active α-satellite SF1, SF2, SF3 and SF01 supra-chromosomal families (Fig. 2e, f). The SF1 family has the slowest methylation rate (0.2x) and reaches a limit at <30% median methylation, indicating that a GATC site located in SF1 α-satellite DNA is inaccessible in a large fraction of cells. We note that sequence variation in α-satellite repeats is such that only some repeats have a GATC site, resulting in varying GATC site densities (Supplementary Fig. 3). Consequently, we are only probing the accessibility of repeats with GATC sites. In summary, centromeric GATC sites are methylated more slowly than other genomic regions in living cells (Fig. 2f). This is particularly true of the active α-satellite repeats (especially the SF1 family), which are associated with CENP-A-containing nucleosomes, and are only partially accessible in vivo.

## TAD boundary chromatin is accessible

The loop organisation of chromatin depends on the CTCF transcription factor/insulator-binding protein and cohesin, which together define TAD boundaries[38,39]. Consequently, CTCF binding is expected to be quite stable, which is consistent with the exceptionally good nucleosome phasing observed on both sides of CTCF binding sites[40,41]. We examined methylation patterns at and around CTCF sites (Fig. 1i). We detected very strong phasing around CTCF motifs in our Dam data, which is out of phase with nucleosome dyads, as expected (cf. the MNase-seq dyad plot, grey profile). Although phasing is strong, the

nucleosomal DNA is still accessible to Dam, since the mean methylation level increases with time, reaching ~85% after 72 h (Fig. 1i; Supplementary Fig. 1g). The plots also imply that the CTCF site is similarly accessible to Dam, since methylation increases to high levels at the motif, even though the motif is in a small trough in the methylation profiles (Fig. 1i). However, we note that the CTCF motif itself does not contain a GATC site and so there are no data at the motif itself; it is therefore unclear whether it is protected from methylation.

## Heterochromatin is accessible in live cells

We asked whether heterochromatin, as defined by specific histone marks, is accessible in living cells using published ChIP-seq data for MCF7 cells[42]. We grouped all GATC sites located in H3K9me3 peaks (constitutive heterochromatin) or in H3K27me3 peaks (facultative heterochromatin) and compared their methylation with GATC sites associated with euchromatin marks (H3K4me1, H3K4me3, H3K27ac and H3K36me3). We observed that GATC sites in euchromatin are methylated faster than those associated with heterochromatin (Fig. 3a; Supplementary Fig. 4a). However, the rate difference is no more than ~2-fold: 1.2x - 1.6x the genomic average for euchromatin; 0.8x and 0.9x for the two heterochromatic states (Fig. 3a; Supplementary Fig. 4a). Most importantly, these heterochromatic regions are fully accessible to Dam.

In a more sophisticated approach, we compared methylation rates in euchromatin and heterochromatin using a 15-state epigenetic ChromHMM model[43], which we derived using the same MCF7 ChIP-seq data. ChromHMM models identify genomic regions associated with various combinations of histone marks. Our ChromHMM model identifies 11 euchromatin states based on the presence of H3K4me1, H3K4me3, H3K27ac and/or H3K36me3 (Fig. 3b). These are: transcription start sites (TSS; state 1), TSS-flanking regions (states 2, 3 and 4), transcriptionally active regions (state 5), weakly active regions (state 6), four types of enhancer (states 7, 8, 9 and 10), and regions with low levels of H3K27ac (state 11). Our model also defines two heterochromatic states: constitutive (H3K9me3; state 13) and Polycomb-repressed (facultative) (H3K27me3; state 14). Some chromatin is in a bivalent state, characterised by the presence of both the active

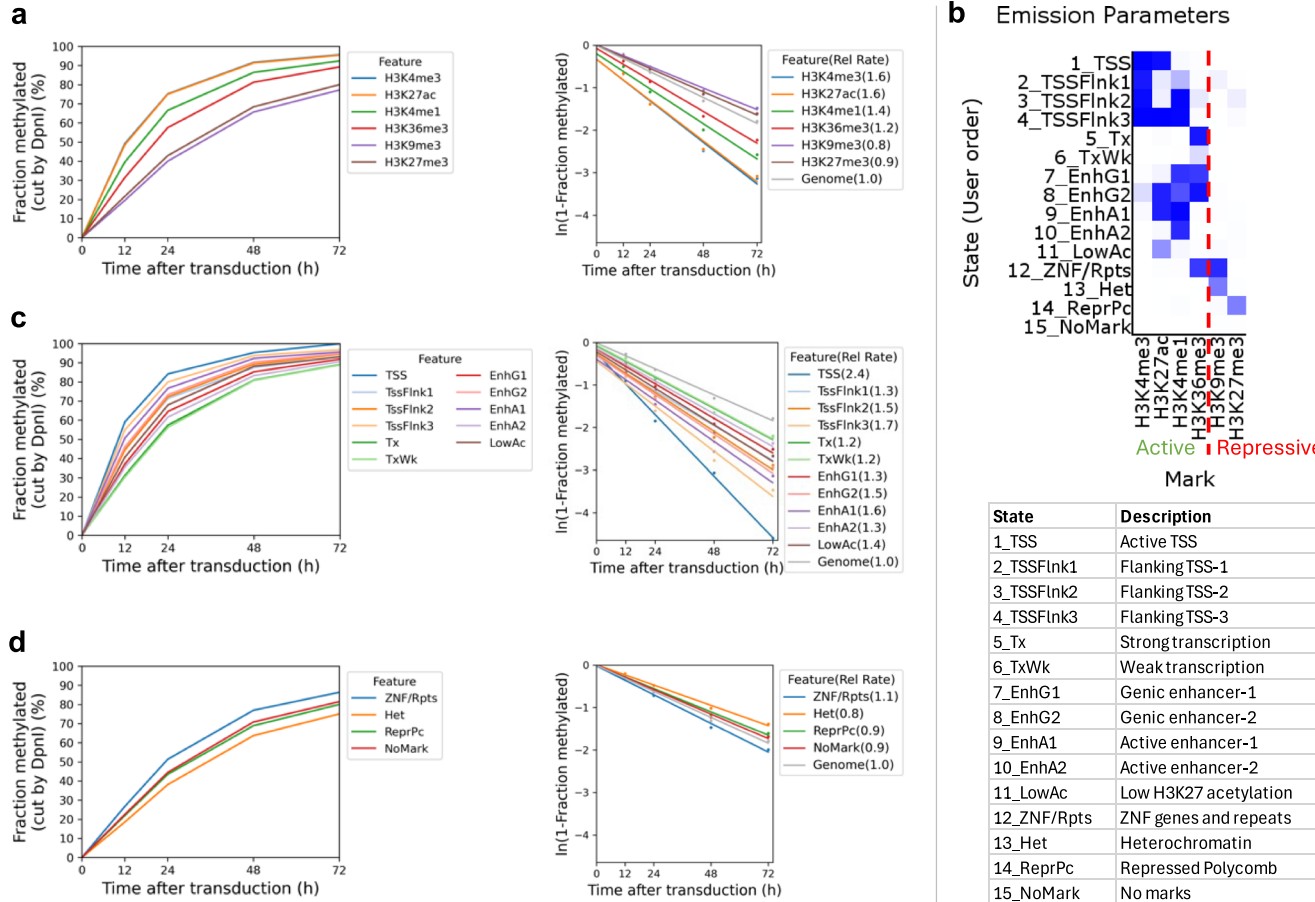

**Fig. 3 | Heterochromatin is accessible but methylated at a slower rate than euchromatin. a** Methylation time courses and methylation rates for the median GATC site in regions with active histone marks (H3K4me1, H3K4me3, H3K27ac, H3K36me3) and inactive histone marks (H3K9me3 and H3K27me3). **b** ChromHMM model defining 15 epigenetic states in MCF7 cells, defined as active or inactive chromatin based on the histone marks. **c** Methylation time courses and methylation rates for the median GATC site in the active chromatin (euchromatin) states defined by the HMM model. **d** Methylation time courses and methylation rates for the median GATC site in the inactive chromatin (heterochromatin) states defined by the HMM model.

H3K36me3 mark and the inactive H3K9me3 mark (state 12). State 15 has none of the histone marks for which we have data and accounts for ~56% of the MCF7 genome. We plotted the methylation kinetics for the active and repressed chromatin states separately for ease of comparison (Fig. 3c,d; Supplementary Fig. 4b). The repressed states are methylated more slowly than the active states, but importantly, even the repressed heterochromatin states trend toward complete methylation (compare Fig. 3c, d).

With the exception of state 1 (TSS), the actual methylation rate differences are not large, ranging from ~1.7x the median genomic rate for most of the euchromatin states to ~0.8x the median genomic rate for the heterochromatin states (compare Fig. 3c, d). The relative methylation rate for GATC sites in state 1 (H3K4me3 and H3K27ac) is relatively high (2.4x and 4.6x for biological replicates), consistent with the inclusion of NDRs in this state (Fig. 3c,d; Supplementary Fig. 4b). We conclude that GATC sites located in heterochromatin are accessible in living cells.

**Accessibility is not due to replication**
We considered the possibility that global genome accessibility in living MCF7 cells might be due to DNA replication. It proved technically challenging to synchronise and maintain MCF7 cells in the G1 phase of the cell cycle prior to transduction. We were also unable to obtain fully confluent MCF7 cells. Consequently, to confirm our findings more broadly and to test the possible role of replication, we performed the same experiment using MCF10A cells, a normal human breast epithelial cell line. Time course experiments after transduction of MCF10A cells with the same Dam-expressing adenovirus produced similar results to those obtained for MCF7 cells (Supplementary Fig. 5). Thus, the genome is also globally accessible in normal MCF10A cells, suggesting that this accessibility is not due to the cancerous nature of MCF7 cells. We grew MCF10A cells to confluence, when the cells cease to replicate their DNA, and then repeated the transduction time course (Supplementary Fig. 6). We observed similarly high accessibility in these arrested cells. We conclude that DNA replication is not a major contributor to genome accessibility.

**The X-chromosome is methylated slowly**
Dosage compensation results in inactivation of one of the two copies of the X-chromosome in female cells and its condensation into heterochromatin (the 'Barr body'; reviewed by ref. [44]). We reasoned that the active X-chromosome would be methylated faster than the inactive X, predicting an intermediate average methylation rate relative to the autosomes, because our method cannot distinguish between the two copies. Since MCF7 cells have aberrant ploidy, whereas MCF10A cells have normal ploidy, we focused our analysis on MCF10A cells. Using the average methylation rate data for 100 kb chromosome windows (Supplementary Fig. 5i), we plotted the fraction of windows with a given relative rate for each of the 23 chromosomes (Supplementary Figs. 5l, 6k). This approach separates out the centromeric chromatin regions, which are methylated more slowly in all chromosomes. We observed that the mean relative methylation rate of the most common

autosomal 100 kb window is 1.05, whereas that of the X-chromosome is 0.85. This result is consistent with slower methylation of the inactive X-chromosome due to its heterochromatic nature.

## Dam expression has little effect on gene expression

We tested whether Dam expression affects gene expression patterns in MCF7 cells by performing RNA-seq before and after treatment with the Dam-GFP expressing adenovirus or a GFP-expressing control adenovirus (Supplementary Fig. 7a). A scatter plot comparing gene expression before and after treatment with the Dam-GFP virus for 72 h shows a Pearson correlation of 0.96, indicating that the virus causes only minor changes in gene expression. A comparison of the Dam-GFP virus with the control GFP virus after 72 h treatment gives a higher correlation (0.99), indicating that some of the minor changes can be attributed to the virus rather than to any effects of Dam methylation. MNase-seq data for untreated and Dam-GFP virus-treated MCF7 cells show that Dam expression has little effect on the chromatin organisation of active and inactive genes and around CTCF motifs (Supplementary Fig. 7b, c). MCF7 cells are ~90% viable after 72 h of Dam-GFP or GFP control virus treatment (acridine orange/propidium iodide staining).

## Limited accessibility in isolated nuclei

It has been shown previously that genome accessibility is limited in nuclei isolated from both yeast and mouse liver cells[27,32,45]. To determine whether this is also true for MCF7, we treated isolated MCF7 nuclei with increasing concentrations of purified Dam enzyme. After a 30 min incubation at 37 °C, genomic DNA was purified and digested with DpnI (Fig. 4a). In comparison with the fully digested unmethylated DNA marker ('MboI' lane), nuclei samples showed incomplete DpnI digestion even at the highest Dam concentration. A clear nucleosome ladder pattern is observed in all of the Dam-treated samples, suggesting that Dam methylates linker DNA, but not nucleosomal DNA in nuclei. This result contrasts with the almost complete methylation observed in living cells (cf. Fig. 1c).

Genomic analysis confirmed that methylation is limited in nuclei (Fig. 4b; Supplementary Fig. 8). The median of all genomic GATC sites reached a plateau at ~38% methylation, indicating that the median GATC site is accessible in ~38% of nuclei and inaccessible in the remaining ~62% of nuclei. The plot shows that there are two different methylation rates in nuclei, unlike in live cells, which show only one rate (Fig. 1d). In nuclei, the initial rate is relatively fast and attributed to methylation of the linker DNA between nucleosomes; the second rate is extremely slow, virtually a plateau, and attributed to dissociation and re-association of the outer ~10 bp of nucleosomal DNA on each side[27,32]. Thus, nucleosomes protect DNA from methylation in nuclei, but not in living cells.

Examination of Dam methylation around the TSS revealed that active genes display improved nucleosome phasing relative to living cells (compare Fig. 4c with Fig. 1e). Methylation of active genes in nuclei reaches a limit at ~45% in the regions flanking the promoter NDR and a limit of ~67% in the promoter NDR (Fig. 4c). In contrast, methylation of inactive genes in nuclei is uniformly limited to ~45% over the entire region, including the inactive promoters, which are not nucleosome-depleted (Fig. 4d; MNase-seq data: grey profile). Methylation is also limited around CTCF motifs in nuclei (Supplementary Fig. 8d). The NDR associated with CTCF motifs is ~70% accessible in nuclei and flanked by well-phased nucleosomes. Euchromatin and heterochromatin regions have very similar accessibilities in nuclei (Supplementary Fig. 4c). Application of our ChromHMM model to the nuclei data revealed that the active states reach a limit methylation of 40% to 50%, except for the TSS state which reaches ~60% (Supplementary Fig. 4d). The TSS state is higher because it includes promoter NDRs. All of the inactive states show almost identical limit median methylation, at 35%-45%, which is slightly lower than the median limit

for the active states. We conclude that accessibility is severely limited in nuclei, unlike in living cells.

Analysis of methylation at the various types of centromeric repeat in nuclei indicated that the limit median methylation ranged from ~45% for non-satellite and other satellite repeats, similar to that observed for gene bodies in nuclei, down to ~25%-35% for HSat1 and active and inactive α-satellites (Supplementary Fig. 9a, b). The active α-satellite SF2, SF3 and SF01 families were methylated to 25%-30% maximum (Supplementary Fig. 9c, d). Furthermore, the α-satellite SF1 family reached a limit methylation at only ~15%, indicating that GATC sites in SF1 repeats are mostly inaccessible in nuclei. The methylation kinetics of these centromeric regions in nuclei are similar to those observed in living cells, which also tend toward a limit (Fig. 2c, e), although the methylation levels reached at α-satellite repeats in living cells are generally higher than in nuclei. Notably, the SF1 α-satellite family is the slowest methylating region observed in vivo (Fig. 2f) and the least accessible in isolated nuclei (Supplementary Fig. 9c, d).

We note that the experimental conditions for Dam methylation in live cells and isolated nuclei should not be compared, because they are not the same type of experiment. In vivo, the Dam concentration increases very slowly with time after transduction (Fig. 1b; Supplementary Fig. 5b) and so does the amount of DNA substrate as the cells continue to divide, whereas in nuclei experiments, the Dam and nuclei concentrations are constant. This is an important distinction because the methylation rate should be directly proportional to the enzyme and substrate concentrations. Furthermore, there is wide variation in the time (in hours) before individual MCF7 cells become GFP-positive (as we observed in yeast cells[32]).

In summary, we propose that nucleosomes are globally dynamic in vivo, resulting in transient exposure of nucleosomal DNA to Dam and eventually in complete methylation, whereas nucleosomes are static in nuclei due to the absence of chromatin remodeling (which requires ATP and regulatory signals), and protect their DNA from methylation (Fig. 4e).

## Discussion

We have measured the accessibility of GATC sites genome-wide in vivo. We expected to find that human euchromatin would resemble yeast chromatin in being globally accessible and this is indeed the case. We proposed that yeast nucleosomes are in continuous flux in living cells, but not in nuclei, where they are static[32]. Such a flux may occur either through nucleosome removal and replacement, or by sliding along the DNA, and/or through reversible conformational changes (Fig. 4e). It is likely that all three mechanisms occur through the agencies of multiple ATP-dependent chromatin remodelers. This ATP-dependent flux renders the underlying DNA open to methylation by Dam and, by inference, to sequence-specific transcription factors.

The general accessibility of both active and inactive genes to Dam suggests that the widely accepted model that inactive genes are inactive because transcription factor binding sites in their promoters are blocked by nucleosomes may no longer be tenable. It seems unlikely that nucleosomes present a permanent block to transcription factor binding, thus maintaining genes in the repressed state. The role of pioneer factors, which have similar affinity for nucleosomal and non-nucleosomal sites, is unclear, given nucleosome flux. However, pioneer factors would be predicted to bind faster, since they do not have to wait for nucleosome dynamics to expose their cognate sites. Pioneer factors might also be important for initiating nucleosome dynamics. Alternatively, gene regulation may occur primarily through regulation of transcription factor gene expression, location (e.g. retention in the cytosol) or activity (e.g. post-translational modifications and allosteric effects), and through non-coding RNA expression.

We also expected that human heterochromatin might either be resistant to Dam methylation, because of its highly condensed state, or that it might be similar to chromatin in nuclei, with immobile

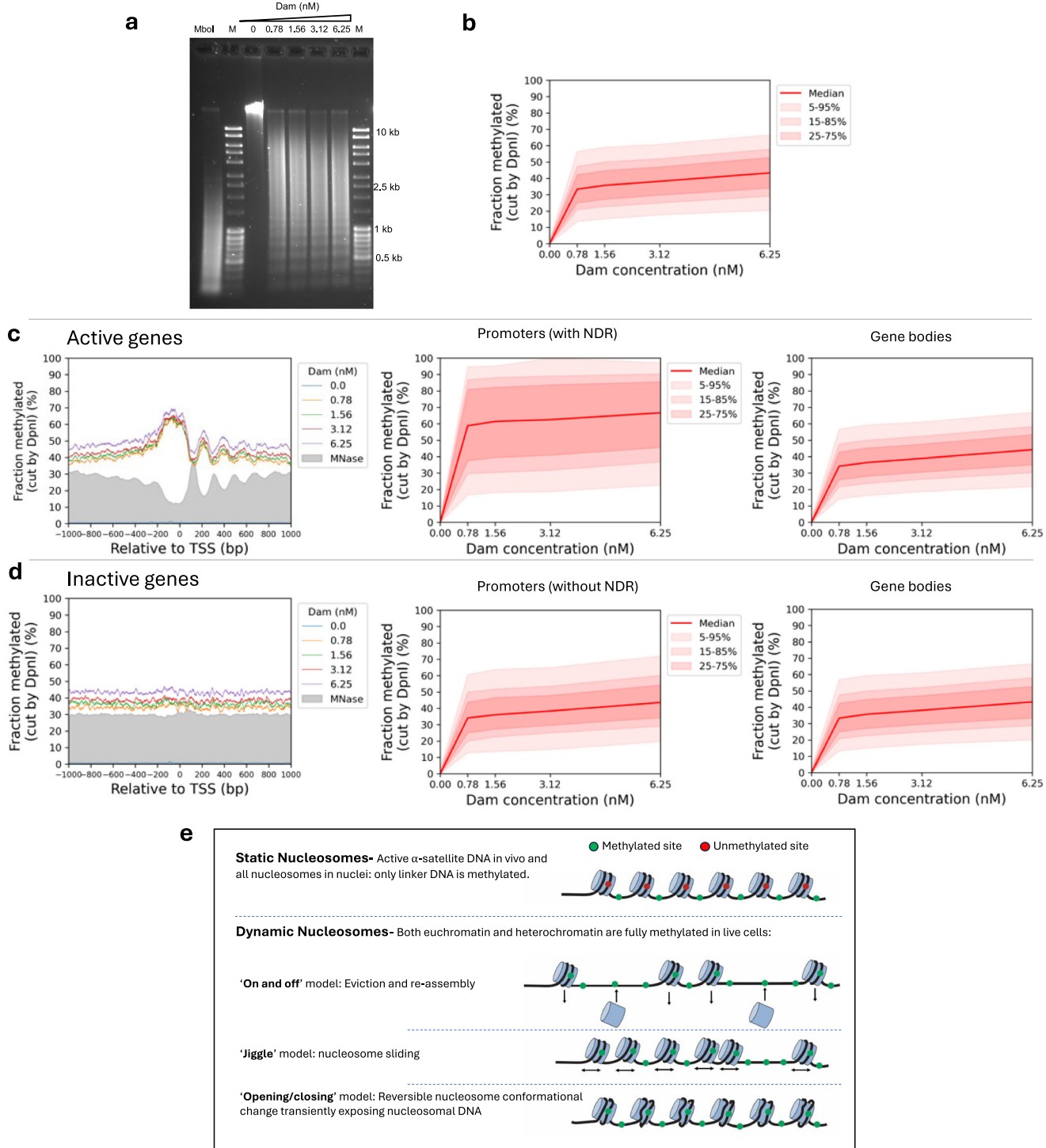

**Fig. 4 | Genome accessibility is limited in isolated MCF7 nuclei. a** Agarose gel electrophoresis of DpnI-digested genomic DNA purified from nuclei treated with increasing amounts of Dam. M, DNA marker. MboI, unmethylated control DNA fully digested at GATC sites by MboI. **b** Methylation of all the GATC sites in the human genome as a function of Dam concentration. Red line: methylation of the median GATC site; shading indicates the data range. **c** Active genes: Nucleosome phasing and methylation of the median GATC site in promoters or gene bodies as a function of Dam concentration. Red line and shading: median GATC site methylation with data range indicated. Grey profile: nucleosome dyad distribution in nuclei (MNase-seq data for MCF7 cells arbitrarily normalised to 30%). **d** The same analysis for inactive genes. **e** Possible ATP-dependent mechanisms for generating accessibility in living cells (based on the known activities of various ATP-dependent chromatin remodelers).

nucleosomes, such that only linkers are methylated. Instead, we observed a trend towards complete methylation, albeit at a somewhat slower rate than for euchromatin. The relatively slow methylation of heterochromatin may be due to a combination of various factors, perhaps including slower nucleosome flux relative to euchromatin, the

highly condensed nature of heterochromatin, and the presence of heterochromatin proteins (e.g., HP1 or Polycomb complexes). Nevertheless, both constitutive and facultative heterochromatin are generally accessible in living human cells. This observation is inconsistent with models proposing that heterochromatin condensation prevents

access to the DNA it contains, resulting in gene repression (discussed by ref. 8). Our data show that heterochromatic DNA is generally accessible and highly dynamic at the nucleosomal level in live cells, unlike in isolated nuclei.

The only genomic regions displaying limited accessibility in living human cells are the centromeric active α-satellite repeats. These elements are methylated slowly relative to other genomic regions and, unlike those regions, reach a limit at 50–60% methylation, which may suggests that centromeres are more densely packed, potentially related to the kinetochore. The SF1 α-satellite repeats are methylated even more slowly, reaching a limit at only ~30% methylation. The nucleosomes in active α-satellite repeats are enriched in centromeric H3 (CENP-A) and so resemble yeast centromeric nucleosomes in their resistance to methylation in vivo[32]. The limited accessibility of centromeric active α-satellite repeats in vivo is similar to that observed for all genomic regions in isolated nuclei. This observation suggests that centromeric chromatin in live cells is static, not dynamic, with little or no nucleosome flux, such that linkers are methylated and nucleosomal DNA is protected (Fig. 4e).

In summary, we have measured the accessibility of the human genome in living cells. We find that the genome is generally accessible at the nucleosomal level, including classical heterochromatin regions marked by H3K9me3 or H3K27me3. The exception is the centromeric active α-satellite repeats, which exhibit limited accessibility. We propose that nucleosome flux creates a genome-wide open chromatin environment, in which the DNA is packaged but still accessible, facilitating the search for cognate sites by sequence-specific transcription factors.

## Methods

### Cell culture
MCF7 cells (ATCC HTB-22) were cultured in RPMI 1640 with L-glutamine (Corning 10-040-CV) supplemented with 10% Fetal Bovine Serum (FBS) (Corning 35-010-CV) and 1% penicillin/ streptomycin (Gibco 15140-148) at 37 °C with 5% $CO_2$/95% air in a humidified incubator. MCF10A cells (ATCC CRL-10317) were cultured in DMEM/F12 (Gibco 11320-033) supplemented with 5% horse serum (Gibco 26050070), 10 μg/ml insulin (Gibco 12585-014), 20 ng/ml epidermal growth factor (Gibco PHG0311), 500 ng/ml hydrocortisone (Sigma H0888), 100 ng/ml cholera toxin (Sigma C80520), 1% penicillin/streptomycin and 2 mM L-glutamine (Gibco 25030149). Confluent MCF10A cells were obtained by culturing in a 12-well plate for 46 h in the same medium and then transduced in medium lacking all growth factors except horse serum.

### Adenovirus transduction and Dam methylation in living cells
The Dam-3HA-eGFP cassette, codon-optimised for mouse, was constructed by gene synthesis (Thermo Fisher GeneArt) (sequence available on request). The Dam expression cassette was expressed from a CMV promoter in an adenovirus vector (human adenovirus type 5 (dE1/E3); Vector Biolabs), packaged, amplified and purified by Vector Biolabs. The final viral yield was $4.2 \times 10^{10}$ plaque-forming units (pfu) per ml, equivalent to approximately $10^{12}$ viral particles per ml. About a million cells were seeded in each well of a 6-well plate one day prior to adenovirus transduction. The next day, the cells were counted and the amount of adenovirus required to achieve a multiplicity of infection (MOI) of 1000 was pre-incubated at 37 °C for 30 min to improve transfer efficiency[46]. The virus was mixed with 300 μl per well of serum-free, antibiotic-free medium (RPMI 1640 for MCF7; DMEM for MCF10A) and incubated for 5–10 min at room temperature. Meanwhile, the culture medium was aspirated from the 6-well plate and 450 μl of complete medium containing FBS was added. The transduction mixture was added to each well, mixed by gently swirling the plate a few times, and incubated at 37 °C for 4 h. Then 1.8 ml of complete culture medium containing serum was added to each well.

Confluent MCF10A cells were treated slightly differently: a 12-well plate and an MOI of 2000 was used (twice as much adenovirus was needed for confluent cells as for log phase cells, possibly because confluent cells are smaller with less surface area exposed to the virus). The virus was mixed with 136 μl DMEM per well and incubated for 5-10 min at room temperature. Meanwhile, the culture medium was aspirated and 204 μl of complete medium containing FBS was added. The transduction mixture was added to each well, mixed and incubated at 37 °C for 8 h. Then 1 ml of complete culture medium containing serum was added to each well.

The time course started at this point, with cells harvested after 12 h, 24 h, 48 h and 72 h (for MCF10A the time points were 12 h, 24 h, 36 h and 48 h). Cells were detached from the well by washing with 1 ml PBS and then incubating in 0.4 ml 0.25% (w/v) trypsin-0.53 mM EDTA solution (ATCC 30-2101) at 37 °C for 5 min (15 min for MCF10A). Next, 1 ml medium was added to the well, cells were collected by centrifugation at 100 g for 5 min, after which the medium was aspirated, and the cells were resuspended in 1 ml medium. The cells were counted, divided up for DNA and protein extraction, quickly frozen on dry ice, and stored at -80 °C. DNA extraction was performed using the PureLink Genomic DNA Mini Kit (Invitrogen) according to the manufacturer's guidelines. Purified genomic DNA (1.2 - 1.5 μg) was digested with 10 units of DpnI (New England Biolabs (NEB) R0176L) in NEB CutSmart buffer for 2 h at 37 °C.

### RNA-seq
About 0.5 million MCF7 cells were transduced using the Ad-Dam-3HA-eGFP adenovirus or with a control eGFP expressing adenovirus vector (Vector Biolabs #1060) at an MOI of 1000, as described above. Cells were harvested before adenovirus treatment and 72 h after transduction with each virus. Cell pellets were immediately resuspended in 0.5 ml Trizol reagent (Invitrogen 15596026), transferred to dry ice and stored at -80 °C. RNA was extracted using the Direct-zol RNA MiniPrep kit (Zymo Research R2051) according to the manufacturer's protocol. RNA concentrations were measured using a Nanodrop spectrophotometer. Purified RNA samples from three experiments were submitted to the NICHD Molecular Genomics Core for library preparation (TruSeq Stranded mRNA kit; Illumina 15031047) and sequencing.

### Immunoblotting
A pellet containing 0.3–0.5 million cells was resuspended in 0.25 to 0.4 ml of 1x lithium dodecyl sulphate buffer (Invitrogen NP0007) supplemented with 0.2 M 2-mercaptoethanol and heated for 5 min at 99 °C; 10 μl was loaded on to each of two 4–12% bis-Tris polyacrylamide gels (Invitrogen NP0336) and run using MOPS/SDS running buffer (Invitrogen). Transfer of proteins to a membrane and signal development with horseradish peroxidase-conjugated anti-HA (3F10; Roche 12013819001) or anti-tubulin (Abcam ab-185067) antibodies were performed as described[32].

### FACS analysis
Propidium iodide staining and flow cytometric DNA analysis of MCF7 and MCF10A cells were performed as follows[47]. Cells (0.1–0.2 million) were resuspended in 50 μl cold buffer (250 mM sucrose, 40 mM tri-sodium citrate, 5% v/v DMSO) and frozen at −80 °C. For FACS, cells were thawed and 200 μl of ice-cold Solution A (0.03 mg/ml trypsin, 3.4 mM trisodium citrate, 0.1% v/v NP-40, 1.5 mM spermine tetrahydrochloride, 0.5 mM Tris-HCl pH 7.6) was added. The mixture was incubated at room temperature for 5 min. Subsequently, 100 μl of ice-cold Solution B (0.5 mg/ml trypsin inhibitor, 0.1 mg/ml RNase A, 3.4 mM trisodium citrate, 0.1% v/v NP-40, 1.5 mM spermine tetrahydrochloride, 0.5 mM Tris-HCl pH 7.6) was added and incubated for another 5 min at room temperature. Finally, 20 μl propidium iodide at 1 mg/ml (Invitrogen P3566) was added and incubated at room

temperature in the dark to prevent photobleaching. The cells were analysed using a FACSCalibur flow cytometer (Becton Dickinson) and Cell Quest Pro software, following the manufacturer's instructions.

## Dam methylation of isolated nuclei

MCF7 cells were cultured in complete medium in a 75 cm² flask and re-passaged into a new flask after 2-3 days of growth. When the cells reached approximately 80% confluency, they were trypsinised and harvested. To extract nuclei, a pellet of 3 - 4 million cells was resuspended in 2 ml Buffer A (15 mM Tris-HCl pH 8.0, 15 mM NaCl, 60 mM KCl, 1.5 mM EDTA, 0.5 mM spermidine, 15 mM 2-mercaptoethanol, and protease inhibitors) with 0.03% NP-40. The mixture was gently but thoroughly mixed by pipetting and incubated on ice for 10 min, inverting the tube 2 or 3 times during the incubation. The lysate was centrifuged at 500 g for 2 min at 4 °C, and the supernatant was removed. The nuclei were washed with 1 ml Buffer A. The nuclei were resuspended in 1 ml Buffer A supplemented with fresh S-adenosylmethionine to 0.5 mM, and divided into five 200 μl aliquots. Dam methyltransferase (NEB M0222B-HC2 at 40 U/μl; 8 μg Dam/ml) was added to the aliquots of nuclei: 0, 25, 50, 100, and 200 units (0, 0.8, 1.6, 3.1, 6.3 nM, respectively), gently mixed, and incubated for 30 min at 37 °C. Genomic DNA was extracted using the PureLink Genomic DNA Kit (Invitrogen 2666617). Finally, 1.2 to 1.5 μg purified genomic DNA was digested with 10 units of DpnI as above. To test whether the spermidine present in Buffer A affects accessibility, the experiment was repeated exactly as above, except Buffer A without spermidine was used (Supplementary Fig. 8e-g).

## MNase-seq

MNase (Worthington LS004798) was dissolved to 10 units/μl in 5 mM Na-phosphate buffer pH 7.0, 0.025 mM CaCl₂, aliquoted out, and stored at −80 °C. Untreated and Dam-3HA-eGFP adenovirus-treated (72 h) MCF7 cells (3 to 4 million) were resuspended in 2 ml Buffer B (15 mM Tris-HCl, pH 8.0, 15 mM NaCl, 60 mM KCl, 1 mM EDTA, 2 mM CaCl₂, 0.5 mM spermidine, 0.03% NP-40, 15 mM 2-mercaptoethanol and protease inhibitors). The cells were gently lysed by pipetting and incubated on ice for 10 min, inverting the tube 2 or 3 times during incubation. The lysate was centrifuged at 500 g for 2 min at 4 °C and the supernatant was removed. The nuclei were washed with 1 ml Buffer B without NP40 and resuspended in 1.3 ml Buffer B without NP40. MNase was added to six tubes of 200 μl nuclei, as follows: 12.5 U, 25 U, 50 U, 100 U, 200 U and 400 U, gently mixed, and incubated for 3 min at 25 °C. MNase-digested DNA was purified using the PureLink Genomic DNA kit (Invitrogen 2666617) and analysed in an agarose gel. For accurate and even nucleosome mapping, we chose digests with a dominant band at ~150 bp corresponding to >80% of the DNA (typically 25 U, 50 U and 100 U), prepared paired-end libraries, and performed low-coverage sequencing to identify the digest with the most optimal DNA fragment length distribution[48]. In the case of untreated cells, the 25 U sample (Replicate 1) and the 50 U sample (Replicate 2) were chosen for high-coverage sequencing; the 25 U samples from adenovirus-treated cells were used for both Replicates 1 and 2.

## Library preparation for paired-end Illumina sequencing

For both nuclei and live cell experiments, DpnI-digested genomic DNA was purified using 1.8 vol. AMPure XP beads (Beckman). Paired-end libraries were prepared as described[32] except for the sonication step, in which the DNA was fragmented using a Covaris ME220 ultrasonicator (350 bp program, peak power, 50 W; duty factor 10%; 1,000 cycles per burst; average power 5; total time 170 s per tube). All sequencing was performed using an Illumina NextSeq 2000 machine.

## Computational analysis of methylated fractions

Dam methylation at each individual GATC site is measured separately. For every GATC site, there is a set of 'fraction methylated' values for the time course, one value for each time point. For analysis of GATC sites in a specific genomic region, we calculate the median of all 'fraction methylated' values for every GATC site in the region at each time point. We developed two packages for methylated fraction analysis: snake-makeMethylFrac and methylFracAnalyzer. SnakemakeMethylFrac, a Snakemake workflow[49], processes raw Illumina paired-end reads to determine methylated fractions at all GATC sites. Bowtie2 v2.5.1[50] is used for alignment and bedtools v2.31.1[51] is used to calculate the occupancy (fragment coverage) and 5'-end counting. GATC sites that overlap with CpG sites were filtered out, because DpnI cannot cut GATm⁵C. GATC half-sites that are within 150 bp of each other were also filtered out because small DNA fragments <150 bp tend to be lost during sample purification. The output includes SQLite database and bigwig files. We use pandas[52], pyBigWig ([https://github.com/deeptools/pyBigWig](https://github.com/deeptools/pyBigWig)), biopython[53], matplotlib[54] and seaborn[55] in this workflow. MethylFracAnalyzer processes bigwig files from Snakema-keMethylFrac for downstream analysis. It calculates percentiles for each feature, methylation rates from median methylated fractions and relative methylation rates. It computes the average methylated fraction in 100-kb windows (T2T v1.1 assembly) and methylation rates using average methylated fractions. It calculates the average methylated fraction relative to the TSS of active and inactive genes, and relative to CTCF sites, smoothed in 21 bp windows. Finally, it generates the associated figures. This software uses pandas, pyBigWig, matplotlib, seaborn and statsmodels[56].

## Analysis of RNA-seq, ATAC-seq and MNase-seq data

We calculated the normalised gene expression in TPM for RNA-seq datasets from the GEO database (GSE201262 for MCF7[34]; GSE237066 for MCF10A[57]) and our own RNA-seq data using salmon v1.10.0[58]. We averaged counts across replicates for each transcript using the GENCODE v43 annotation of the Hg38 assembly[59]. For multi-transcript genes, we selected the most highly expressed transcripts and used their TSSs. We used published bigwig files for ATAC-seq datasets for MCF7 (GSE201262[34]) and MCF10A (GSE152410 [60]). ATAC-seq data for 1 kb regions flanking TSSs were extracted; active genes were assigned based on average signal (> 0.4 for MCF7 and >2 for MCF10A). Our MNase-seq data were aligned using Bowtie2 v2.5.1. We selected read fragments for single nucleosomes (fragment length: 120–180 bp) and counted the nucleosome dyads in 2010-bp regions flanking gene TSSs. We normalised dyad counts per gene using average dyad counts per flanking region, then averaged dyad counts across all active and inactive genes, smoothing in 21-bp windows.

## CTCF sites and ChromHMM

CTCF narrowPeaks were obtained from ENCODE[61,62] (ENCSR000AHD for MCF7 and ENCSR193SZD for MCF10A). CTCF motifs were predicted using HOMER v4.11.1[63]. For CTCF phasing analysis, the peaks containing a single copy of the highest frequency motif were selected. For ChromHMM analysis, we used ChIP-seq data from GSE85158 for both MCF7 and MCF10A cells[42], processed with the ENCODE ChIP-seq pipeline v2 ([https://github.com/ENCODE-DCC/chip-seq-pipeline2](https://github.com/ENCODE-DCC/chip-seq-pipeline2)). ChromHMM v1.23 predicted 15 chromatin states using the T2T v1.1 assembly[43]. JHU RefSeqv110 + Liftoff v5.1 annotation ([https://github.com/marbl/CHM13](https://github.com/marbl/CHM13)) were used for feature enrichment. Final chromatin state annotations are available in our GitHub repository ([https://github.com/zhuweix/methylFracAnalyzer](https://github.com/zhuweix/methylFracAnalyzer)).

## Statistics and Reproducibility

Two biological replicate experiments were performed. The panels shown in each main figure belong to the same experiment. Supplementary Figs. generally show the results from both replicate experiments, except for immunoblots, micrographs and DNA gels. The immunoblots and DNA gel analyses were similar in both experiments.

Correlations between biological replicates at the chromosomal level are presented in Supplementary Fig. 10.

## Reporting summary

Further information on research design is available in the Nature Portfolio Reporting Summary linked to this article.

## Data availability

The Illumina sequence data generated in this study have been deposited in the GEO database under the following accession codes: GSE282872 (Methylation data for MCF10A cells). GSE282873 (Methylation data for MCF7 cells). GSE282874 (MNase-seq data for MCF7 cells before adenovirus treatment). GSE292647 (RNA-seq data for MCF7 cells). GSE292648 (Methylation data for MCF7 nuclei (no spermidine)). GSE292649 (MNase-seq data for MCF7 cells after adenovirus treatment). Previously published data used in this study are available at the GEO database under the following accession codes: GSE201262 (MCF7 ATAC-seq and RNA-seq data). GSE237066 (MCF10A RNA-seq data). GSE152410 (MCF10A ATAC-seq data). GSE85158 (MCF7 and MCF10A ChIP-seq data). Source data are provided with this paper.

## Code availability

The code used to analyse the data is available at FigShare and Github: Snakemake workflow to process the Illumina reads for methylated fraction analysis: https://doi.org/10.6084/m9.figshare.26236745. https://github.com/zhuweix/snakemakeMethylFrac. Downstream analysis of methylated fraction from the snakemake pipeline: https://doi.org/10.6084/m9.figshare.26236757. https://github.com/zhuweix/methylFracAnalyzer

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

## Acknowledgements

We thank Alex Vassilev for help with the FACS analysis and the NICHD Molecular Genomics Core Facility for RNA sequencing. This study utilised the high-performance computational capabilities of the Biowulf Linux cluster at the National Institutes of Health (NIH). This research was supported by the Intramural Research Program of the NIH (NICHD).

## Author contributions

H.P. and P.E. performed the experiments; Z.X. performed the bioinformatic analysis; H.P. and D.C. wrote the manuscript.

## Funding

## Competing interests

The authors declare no competing interests.
