## [Transparent Peer Review file · Nature Communications]

Nucleosome dynamics render heterochromatin accessible in living human cells

Corresponding Author: Dr David Clark

Version 0:

Reviewer comments:

Reviewer #1

(Remarks to the Author)

Key results

Prajapati et al. have devised an interesting manuscript, weaving together a novel perspective on chromatin dynamics. Where previously constitutive heterochromatin was thought to be “impenetrable”, rendering regions inaccessible to transcription factors and other transcriptional machinery, the authors suggest that this state is in fact accessible to Dam in living cells. With this they highlight a potential caveat in existing studies that yield data from isolated nuclei, implying that they may not be wholly representative of in vivo contexts. They take a thorough approach, combining numerous datasets to explore the potential of their research question and reveal that whilst all regions of the genome are accessible by Dam, centromeres are methylated at a lower rate than other regions. Equally, they explore the rate of methylation at constitutive and facultative chromatin regions, consider post-translational modifications and compare with isolated nuclei data.

Most important findings:

- Dam can access all chromatin in the MCF7 human cell line (heterochromatin takes longer and α -satellite chromatin is partially inaccessible). This is a very important finding that changes our perception of heterochromatin in human cells, in that much of it can be accessed to varying degrees.
- Facultative chromatin is still relatively accessible (much more than heterochromatin)
- This is (potentially) different to what is seen with isolated nuclei (ATAC and Dam experiments)

Queries/concerns:

- There is concern that expressing Dam at high levels could impact on gene expression and possibly chromatin structure, providing an altered view of the chromatin accessibility. In *Drosophila*, excessive Dam expression causes developmental defects and lethality, and levels have to be controlled through reduced translation (Southall et al., 2013). We strongly recommend performing RNA-seq and ATAC on cells with and without Dam expression. This will determine if Dam expression is causing accessibility and expression changes that could change the interpretation of the results.

- Isolated nuclei experiments – how well can purified Dam protein enter the nuclei? It may just be present at much lower levels than in the live intracellular expression experiments, and hence less methylation. Also, 30 mins Dam vs 72 hrs in human cells, raises questions about how comparable the data is.

- If this is a legitimate finding, why would isolated nuclei be different? This should be discussed more.

Minor queries/concerns:

Result 1: Global accessibility in live human cells

- Which time-point was used for the Mbol experiment (Fig.1C) – would you see the same or different trend at other time points?
- The ATAC data that was used for comparison, was that from a comparable line?
- What about H3K9me chromatin at non-centromeric sites (e.g. developmental genes repressed by H3K9me)? Are differences seen there?
- α -satellite SF1 methylation is < 30% median methylation – is that a difference between different subpopulations of cells? Or

just a proportion of chromatin that is completely inaccessible across all cells?

-

Result 3: TAD boundary chromatin is accessible

- Investigating TAD boundaries i.e. chromatin looping and 3d genome topology. What was the rationale for looking at this?
- Sudden change in referencing format? Vancouver to Harvard? (Fu et al., 2008; Wiechens et al., 2016).
- What does “exceptionally good nucleosome phasing” mean (TAD boundary section) – this is a bit vague - recommend changing the wording.
- The fact that the CTCF motif has very high Dam signal but does not contain a GATC site is a bit concerning? Again, is the Dam-signal saturated? Perhaps due to chromatin looping? – Should be addressed in the text more.

-

Result 4: Heterochromatin is accessible in live cells

- Should highlight and discuss more regarding the differences between profiling live and cells and fixed cells (i.e. fixed cells are a snapshot) and the Dam profiling is cumulative.

Result 5: Not due to replication

- Switch from MCF7 to MCF10A – why not use throughout paper if MCF7 is prone to ploidy, does that not affect analysis and reproducibility?

Result 7: Limited accessibility in isolated nuclei

- Why do they think that accessibility is limited in nuclei compared to living cells? – this should be discussed more.

General comments:

We think this is an exciting and important story that, once concerns are addressed, would be very suitable for publication in Nature Communications. As well as addressing technical concerns, we think that it would benefit from more discussion (either at the end of each section, or include a Discussion section).

(Remarks on code availability)

The code is publically available on Figshare.com. It has text that would be in a readme file. Github might be a more appropriate location for sharing this code.

Reviewer #2

(Remarks to the Author)

In this manuscript, “Nucleosome dynamics render heterochromatin accessible in living human cells,” Prajapati and colleagues describe how living human cells exhibit accessibility behaviors distinct from isolated nuclei. Coupled with their previous work in yeast, the authors show that nucleosome dynamics contribute to accessibility in living cells across species by now using human breast cancer and normal breast cell models. The authors show that chromatin is entirely accessible to Dam methylase (euchromatin more than heterochromatin), with the exception of the centromere. Notably, the authors demonstrate that this observed accessibility is independent of cellular replication. Overall, the results of this manuscript are of significant interest to the field, highlighting the importance of ATP-dependent processes in chromatin structure. Additionally, the study underscores the unique composition of the centromere, warranting further investigation. I recommend this manuscript for publication in Nature Communications, pending the resolution of the following minor concerns:

- 1) One of the most interesting, yet understandable, conclusions from the papers is the finding that there is that (1) centromeric DNA is unable to achieve maximal GATC methylation and (2) there are varying degrees of enzymatic activity within certain centromeric regions. This suggests the presence of densely-packed centromere potentially related to the kinetochore. This model is consistent with observations that have identified a methylation dip within the centromere, as seen in PMID: 39657946. In this regard, the authors have identified SF1 to be the most resistant to this affect among the subfamilies, and it would be interesting to hear what the authors would account for this observation.
- 2) The study does an excellent job conveying their analysis to the reader as it is both clearly described and intuitive. When investigating rates, the authors plot % methylation vs time as well as a transformation of $\ln(1 - \% \text{ affected})$ vs. time. The linear observation of the rate after transformation is consistent with first-order kinetics across all features. However, it does not appear this calculation is normalizing to feature size (where size is equal the base pair length of a given genomic feature). This is necessary to ensure fair comparisons, as smaller features may appear to react faster simply due to having fewer available sites rather than an intrinsic enzymatic preference. This appears to hold true as the expected euchromatin regulatory elements chromatin rates have a steeper slope compared to the gene body associated euchromatin. In order to represent the data fairly, feature size should be accounted for in their analysis.
- 3) One of the limitations in the analysis described is in the method starting at Line 574: “GATC sites that overlap with CpG sites were filtered out, because DpnI cannot cut GATm5C. GATC half-sites that are within 150 bp of each other were also filtered out because small DNA fragments < 150 bp tend to be lost during sample purification.” I have questions about this and how it relates to two types of observations in the data presented: (1) In Figure 1h, the % methylated reported includes the CpG island feature. I expect similar behavior to the promoter as it is shown, but how does filtering affect this specific feature? (2) Furthermore, regarding the centromere analysis where I expect many repeats, what is the relative contribution of GATC repeat-containing sites that overlap with CpG dinucleotides? Could a disproportionately high level of CpG methylation in these regions contribute to the lack of methylated positions?
- 4) There is no mention of telomeres within the study or discussion which I suspect is due to the lack of interpretable data from these regions. There would be added value to the discussion based on the author's insight into this genomic feature, building on their understanding of chromatin accessibility in living human cells, as informed by this study.

(Remarks on code availability)

Reviewer #3

(Remarks to the Author)

(Remarks on code availability)

I did not review the code.

Reviewer #4

(Remarks to the Author)

(Remarks on code availability)

Reviewer #5

(Remarks to the Author)

How chromatin organization modulates DNA accessibility for gene regulation remains an important question in cell biology. Prajapati et al. utilized DNA adenine methyltransferase to address this question and investigated genome accessibility at all GATC sites in living MCF7 and MCF10A cells. The authors found the human genome was globally accessible in living cells and that heterochromatic GATC sites were methylated only more slowly than euchromatic sites. Interestingly, the centromeric sites were partly inaccessible. The authors' approach is unique, and their findings are exciting and valuable to a wide range of researchers. For publication in Nature Communications, several issues need to be addressed. My specific comments are as follows:

Major points:

(1) Figs. 1 and 2. The results are very impressive. However, I wonder how the frequencies (densities) of GATC sites differ among the analyzed regions, such as active gene regions (promoters and gene bodies), inactive gene regions (promoters and gene bodies), and other regions shown in panel h. Since the GATC frequency in each region may affect the results and their interpretation, a detailed comparison of these densities would be valuable.

For example, the consensus sequence for the 171-bp alpha-satellite does not contain a GATC motif (PMID 2030938), suggesting that GATC sites are very rare in alpha-satellite regions.

(2) Experimental conditions of live cells and isolated nuclei seem very different and are not comparable. To accurately compare methylation kinetics, the authors should quantify Dam expression levels in the cells.

Furthermore, the nuclei isolation buffer contains spermidine, which can induce chromatin compaction. The results cannot exclude the possibility that the limited accessibility was due to the artificial compaction during the nuclei preparation.

(3) Line 491. The numbers of virus particles (MOI) used for cycling and confluent MCF10A cells were quite different. Could we compare the methylation kinetics in cycling/confluent MCF10A data? The authors need comparable expression levels of the Dam.

(4) Fig. 4. I consider how Dam can access GATC sites in the genome very intriguing. Can nucleosomes block methylation of nucleosomal DNA except for linker DNA? It is known that H2A and H2B in the nucleosome are frequently replaced (PMID: 11425866). I wonder whether Dam could methylate nucleosomal DNA with only H3 and H4.

(5) After extensive methylation of the genomic DNA, what impact did this have on chromatin structure and organization? Additionally, were the cells still viable following up to 72 hours of Dam expression?

(6) Line 93. Unlike the yeast genome, which is fully methylated within several hours (Ref. 30), it took 48-72 hrs for human genomic DNA. Why? Is this because human genome chromatin is generally more condensed than yeast chromatin (PMID: 37385880)?

Minor points:

(1) Lines 19-20 (also related to Line 329). "euchromatin, which is relatively decondensed... and heterochromatin, which is condensed..."

Recent studies have suggested that euchromatin can also be condensed, with only limited regions, such as enhancers and

transcription start sites (TSSs) remaining decondensed (PMIDs: 32967822, 37018405, 37385880). It would be beneficial to include this updated perspective on chromatin structure.

(2) Lines 68 and 305. To recruit ATP-dependent chromatin remodelers, local nucleosome dynamics in euchromatin and heterochromatin may be important. It would be valuable to discuss the nucleosome fluctuations in these regions (bioRxiv, 2024.10.20.618801; PMID: 32574554).

(Remarks on code availability)

Version 1:

Reviewer comments:

Reviewer #1

(Remarks to the Author)

The authors have addressed my concerns very comprehensively and I highly recommend for acceptance and publication.

(Remarks on code availability)

Reviewer #2

(Remarks to the Author)

The authors have done an excellent job of addressing the reviewers' concerns and I support publication.

(Remarks on code availability)

Reviewer #3

(Remarks to the Author)

(Remarks on code availability)

Reviewer #4

(Remarks to the Author)

(Remarks on code availability)

Reviewer #5

(Remarks to the Author)

The authors addressed most of my previous comments, and the revised manuscript is clearly improved. However, prior to publication in Nature Communications, there are a couple of remaining issues that still need to be addressed:

Comment 1:

While the authors presented the methylated fraction data, they should also provide data normalized by GATC site density, as this is essential for a fair comparison across genomic regions. This point was also raised by Reviewer #2.

Comment 3:

To support the comparison of methylation kinetics between cycling and confluent MCF10A cells, the authors should directly compare Dam expression levels in these two conditions by western blot. Without confirming comparable expression levels, interpretation of kinetic differences remains uncertain.

(Remarks on code availability)

Response to Reviewers

We thank the Reviewers for their very thoughtful and constructive reviews. Our responses are in blue text. Text changes in the manuscript are also in blue text (some changes are to conform to *Nature Communications* formatting).

Please note that on page 4, line 160, the methylation rate for centromeres in Replicate 1 has been corrected from 0.3 to 0.4 (Replicate 2 is unchanged at 0.4) to include the contribution of the X-chromosome, which was inadvertently excluded in the original calculation. The plots in Supplementary Fig. 1f have also been corrected.

Reviewer #1 (Remarks to the Author)

Key results

Prajapati et al. have devised an interesting manuscript, weaving together a novel perspective on chromatin dynamics. Where previously constitutive heterochromatin was thought to be “impenetrable”, rendering regions inaccessible to transcription factors and other transcriptional machinery, the authors suggest that this state is in fact accessible to Dam in living cells. With this they highlight a potential caveat in existing studies that yield data from isolated nuclei, implying that they may not be wholly representative of in vivo contexts. They take a thorough approach, combining numerous datasets to explore the potential of their research question and reveal that whilst all regions of the genome are accessible by Dam, centromeres are methylated at a lower rate than other regions. Equally, they explore the rate of methylation at constitutive and facultative chromatin regions, consider post-translational modifications and compare with isolated nuclei data.

Most important findings:

- Dam can access all chromatin in the MCF7 human cell line (heterochromatin takes longer and α -satellite chromatin is partially inaccessible). This is a very important finding that changes our perception of heterochromatin in human cells, in that much of it can be accessed to varying degrees.
- Facultative chromatin is still relatively accessible (much more than heterochromatin)
- This is (potentially) different to what is seen with isolated nuclei (ATAC and Dam experiments)

Queries/concerns:

- There is concern that expressing Dam at high levels could impact on gene expression and possibly chromatin structure, providing an altered view of the chromatin accessibility. In *Drosophila*, excessive Dam expression causes developmental defects and lethality, and levels have to be controlled through reduced translation (Southall et al., 2013). We strongly recommend performing RNA-seq and ATAC on cells with and without Dam expression. This will determine if Dam

expression is causing accessibility and expression changes that could change the interpretation of the results.

RESPONSE: As suggested, we have performed RNA-seq before and after virus treatment, including a virus control that expresses GFP but not Dam (the original virus expresses a Dam-GFP fusion). These data are presented in new Supplementary Fig. 7a and described at the top of page 7. We find that gene expression changes due to Dam expression are very limited, as shown by the excellent Pearson correlations. We performed MNase-seq instead of the suggested ATAC-seq because Reviewer 5 raised a similar point, asking whether chromatin organisation is affected by Dam methylation. We compared MNase-seq data for Dam adenovirus-treated cells with MNase-seq data for untreated cells. We found little difference in the overall chromatin organisation of active and inactive genes before and after virus treatment (new Supplementary Fig. 7b). MCF7 and MCF10A cells are ~90% viable after 72 h of treatment with the Dam-GFP virus or the GFP control virus, as shown by acridine orange/propidium iodide staining.

- Isolated nuclei experiments – how well can purified Dam protein enter the nuclei? It may just be present at much lower levels than in the live intracellular expression experiments, and hence less methylation. Also, 30 mins Dam vs 72 hrs in human cells, raises questions about how comparable the data is.

RESPONSE: We know that Dam enters practically all of the nuclei because there is very little undigested (unmethylated) genomic DNA in the gel (Fig. 4a). If the limiting factor for methylation is the Dam concentration in the nuclei, the fraction methylated would continue to increase with increasing Dam concentration, but it reaches a limit, even though we keep doubling the amount of enzyme added (Fig. 4b-d).

The methylation rates in living cells and isolated nuclei should not be compared, because they are not the same type of experiment. In vivo, the Dam concentration is changing with time (Fig. 1b) and so is the amount of DNA substrate as the cells continue to divide, whereas in nuclei experiments, the Dam and nuclei concentrations are constant. This is an important distinction because the methylation rate should be directly proportional to the enzyme and substrate concentrations. Furthermore, we observe a wide variation in the time (in hours) before individual MCF7 cells become GFP-positive (as we observed in yeast cells).

In principle, we could add much less Dam to the nuclei and do a time course over 72 h, similar to the virus treatment (although the enzyme concentration would still be constant). However, long time courses should be avoided because of the risk of proteolysis, enzyme inactivation and SAM decomposition. Consequently, we elected to do a Dam titration for a fixed short time (30 min) in the presence of protease inhibitors.

We have inserted an explanatory paragraph at the end of the Results on page 8.

- If this is a legitimate finding, why would isolated nuclei be different? This should be discussed more.

RESPONSE: We did not discuss the nuclei experiment in enough detail. We are proposing that isolated nuclei are different because the ATP-dependent remodelers are non-functional in the

absence of ATP and other signals (see the first paragraph in the Discussion on page 8 and Fig. 4e). To clarify, we have added the following paragraph to the text on page 7:

"The plot shows that there are two different methylation rates in nuclei, unlike in live cells, which show only one rate (Fig. 1d). In nuclei, the initial rate is relatively fast and attributed to methylation of the linker DNA between nucleosomes; the second rate is extremely slow, virtually a plateau, and attributed to dissociation and re-association of the outer ~10 bp of nucleosomal DNA on each side. Thus, nucleosomes protect DNA from methylation in nuclei, but not in living cells."

And at the end of the Results (page 8): "In summary, we propose that nucleosomes are globally dynamic in vivo, resulting in transient exposure of nucleosomal DNA to Dam and eventually in complete methylation, whereas nucleosomes are static in nuclei due to the absence of chromatin remodeling (which requires ATP and regulatory signals), and protect their DNA from methylation (Fig. 4e)."

Minor queries/concerns:

Result 1: Global accessibility in live human cells

- Which time-point was used for the Mbol experiment (Fig.1C) – would you see the same or different trend at other time points?

RESPONSE: The Mbol lane is purified unmethylated genomic DNA that was completely digested with Mbol in vitro. It is just a gel marker for complete digestion by DpnI (equals complete methylation by Dam). Mbol and DpnI both cut at GATC. We incorrectly referred to it as a control. We now use the term "marker" instead of "control" on page 7 and in the legend to Fig. 1c.

- The ATAC data that was used for comparison, was that from a comparable line?

RESPONSE: The ATAC-seq data are for MCF7 cells from another lab. On page 3 (line 118), we state: "Analysis of published ATAC-seq data for MCF7 cells ...".

- What about H3K9me chromatin at non-centromeric sites (e.g. developmental genes repressed by H3K9me)? Are differences seen there?

RESPONSE: There are very few H3K9me3 peaks at centromeres in MCF7 cells. Most H3K9me3 peaks are located in SINEs and LINEs. The inactive genes (developmentally repressed genes) have isolated H3K9me3 peaks but are mostly free of H3K9me3. We find that GATC sites within H3K9me3 peaks inside genes are methylated at a similar rate to other heterochromatic regions (please see figure below).

- α -satellite SF1 methylation is < 30% median methylation – is that a difference between different subpopulations of cells? Or just a proportion of chromatin that is completely inaccessible across all cells?

RESPONSE: We cannot distinguish between these two possibilities because SF1 elements are repeated sequences. We can only say that a limit of < 30% methylation at SF1 elements indicates that the median GATC site in these elements is protected from methylation (inaccessible) in > 70% of cells. We argue that the nucleosomes are static in SF1 chromatin (as they are in nuclei), with the median GATC site present in the linker in < 30% of cells and protected by a nucleosome in > 70% of cells.

Result 3: TAD boundary chromatin is accessible

- Investigating TAD boundaries i.e. chromatin looping and 3d genome topology. What was the rationale for looking at this?

RESPONSE: There is a lot of interest in CTCF sites and their role in defining TAD boundaries. It seemed possible that loop stabilisation may involve reduced nucleosome flux at TAD boundaries, but we don't see that.

- Sudden change in referencing format? Vancouver to Harvard? (Fu et al., 2008; Wiechens et al., 2016). RESPONSE: Fixed.

- What does “exceptionally good nucleosome phasing” mean (TAD boundary section) – this is a bit vague - recommend changing the wording.

RESPONSE: The degree of nucleosome phasing is given by the amplitude of the signal (peak to midline); a flat line indicates no phasing at all (as observed for inactive genes). Larger amplitudes indicate better phasing - please compare the phasing for active genes and CTCF sites in new Supplementary Fig. 7b, c).

- The fact that the CTCF motif has very high Dam signal but does not contain a GATC site is a bit concerning? Again, is the Dam-signal saturated? Perhaps due to chromatin looping? – Should be addressed in the text more.

RESPONSE: The plots involve "joining the dots" corresponding to the GATC site distribution and then smoothing with a 21-bp window. Since there is no GATC site in the CTCF motif and the data are aligned on the motif, there is no data at the motif itself; the points corresponding to GATC sites on each side of the motif are joined and smoothed. We have inserted a clarifying phrase on page 5: "and so there are no data at the motif itself".

Result 4: Heterochromatin is accessible in live cells

- Should highlight and discuss more regarding the differences between profiling live and cells and fixed cells (i.e. fixed cells are a snapshot) and the Dam profiling is cumulative.

RESPONSE: We are using published ChIP-seq data derived from formaldehyde-fixed cells to identify heterochromatin regions. We agree that such data represent a snapshot of events in vivo.

Dam profiling is cumulative as stated by the Reviewer, but we are not sure why this distinction could be important here.

Result 5: Not due to replication

- Switch from MCF7 to MCF10A – why not use throughout paper if MCF7 is prone to ploidy, does that not affect analysis and reproducibility?

RESPONSE: We used MCF7 cells because of the availability of ENCODE data to build the HMM model. There are less data available for MCF10A cells. We used MCF10A cells for the replication test experiment because it's difficult to arrest MCF7 cells (they show limited contact inhibition). Except for this experiment, we did all the experiments on both cell lines to show that a more normal cell line allows the same conclusions. Our biological replicate experiments are in close agreement, so analysis and reproducibility don't seem to be affected by potential ploidy issues.

Result 7: Limited accessibility in isolated nuclei

- Why do they think that accessibility is limited in nuclei compared to living cells? – this should be discussed more.

RESPONSE: As stated above, we did not discuss the nuclei experiment in enough detail. We are proposing that isolated nuclei are different because the ATP-dependent remodelers are non-functional in the absence of ATP and other signals. These remodelers can move, remove or open up nucleosomes in vitro. In their absence, nucleosomes are very stable and protect their DNA from restriction enzymes and DNA methylases. Consequently, we argue that nucleosomes are static in nuclei, such that linkers are fully methylated and nucleosomal DNA is fully protected (this is the basis of the Fiber-seq technique for mapping nucleosome positions in nuclei). The surprise is that we have found that this is not true in vivo, leading us to propose that the remodelers are responsible for a general nucleosome flux that transiently exposes the DNA to Dam, resulting in complete methylation. We clarify these points in the new paragraphs on pages 7 and 8, mentioned above, and in the first paragraph of the Discussion (see Fig. 4e).

General comments:

We think this is an exciting and important story that, once concerns are addressed, would be very suitable for publication in Nature Communications. As well as addressing technical concerns, we think that it would benefit from more discussion (either at the end of each section, or include a Discussion section).(Remarks on code availability)

RESPONSE: We thank the Reviewers for their thoughtful and constructive comments.

The code is publically available on Figshare.com. It has has text that would be in a readme file. Github might be a more appropriate location for sharing this code.

RESPONSE: We agree. We have deposited the code at GitHub also, but there doesn't seem to be a way to provide reviewer access without making it public. We will make it public when our paper is accepted. Currently, the code is available to the Reviewers on FigShare.

Reviewer #2 (Remarks to the Author)

In this manuscript, “Nucleosome dynamics render heterochromatin accessible in living human cells,” Prajapati and colleagues describe how living human cells exhibit accessibility behaviors distinct from isolated nuclei. Coupled with their previous work in yeast, the authors show that nucleosome dynamics contribute to accessibility in living cells across species by now using human breast cancer and normal breast cell models. The authors show that chromatin is entirely accessible to Dam methylase (euchromatin more than heterochromatin), with the exception of the centromere. Notably, the authors demonstrate that this observed accessibility is independent of cellular replication. Overall, the results of this manuscript are of significant interest to the field, highlighting the importance of ATP-dependent processes in chromatin structure. Additionally, the study underscores the unique composition of the centromere, warranting further investigation. I recommend this manuscript for publication in Nature Communications, pending the resolution of the following minor concerns:

1) One of the most interesting, yet understandable, conclusions from the papers is the finding that there is that (1) centromeric DNA is unable to achieve maximal GATC methylation and (2) there are varying degrees of enzymatic activity within certain centromeric regions. This suggests the presence of densely-packed centromere potentially related to the kinetochore. This model is consistent with observations that have identified a methylation dip within the centromere, as seen in PMID: 39657946. In this regard, the authors have identified SF1 to be the most resistant to this affect among the subfamilies, and it would be interesting to hear what the authors would account for this observation.

RESPONSE: We agree with the Reviewer that our data may suggest the presence of densely packed centromere potentially related to the kinetochore. We have added a comment in the Discussion (page 9, line 390). We find that the centromeric dip regions (CDRs) give virtually identical median plots to those shown in Fig. 2e. That is, the variation in SF repeats is also seen in the CDRs and so the methylation kinetics are not the same for all CDRs (please see figure). Therefore we don't think that the CDR is responsible for the differences between the SF repeats. However, the higher resolution of a methylase with more target sites combined with long-read sequencing is probably needed to test this possibility properly.

Dam methylation at CDRs and CENP-A sites in SF1, SF2, SF3 and SF01 repeats in MCF7 cells

2) The study does an excellent job conveying their analysis to the reader as it is both clearly described and intuitive. When investigating rates, the authors plot % methylation vs time as well as a transformation of $\ln(1 - \% \text{ affected})$ vs. time. The linear observation of the rate after transformation is consistent with first-order kinetics across all features. However, it does not

appear this calculation is normalizing to feature size (where size is equal the base pair length of a given genomic feature). This is necessary to ensure fair comparisons, as smaller features may appear to react faster simply due to having fewer available sites rather than an intrinsic enzymatic preference. This appears to hold true as the expected euchromatin regulatory elements chromatin rates have a steeper slope compared to the gene body associated euchromatin. In order to represent the data fairly, feature size should be accounted for in their analysis.

RESPONSE: We don't think normalizing to feature size is appropriate, because we are assaying individual GATC sites. The methylated fraction for each individual GATC site is computed separately for each time point and then the median methylated fraction and ranges are calculated for each group of sites at that time point. The rates cited are for the median GATC site. We argue that euchromatic regulatory elements have a steeper slope because they are nucleosome-depleted, and so Dam does not have to wait for a nucleosome to be moved away or opened up. We note that the feature sizes and GATC site densities are in new Supplementary Fig. 3.

3) One of the limitations in the analysis described is in the method starting at Line 574: "GATC sites that overlap with CpG sites were filtered out, because DpnI cannot cut GATm5C. GATC half-sites that are within 150 bp of each other were also filtered out because small DNA fragments < 150 bp tend to be lost during sample purification." I have questions about this and how it relates to two types of observations in the data presented: (1) In Figure 1h, the % methylated reported includes the CpG island feature. I expect similar behavior to the promoter as it is shown, but how does filtering affect this specific feature? (2) Furthermore, regarding the centromere analysis where I expect many repeats, what is the relative contribution of GATC repeat-containing sites that overlap with CpG dinucleotides? Could a disproportionately high level of CpG methylation in these regions contribute to the lack of methylated positions?

RESPONSE: (1) Regarding CpG islands, there are 40163 GATC sites in CpG islands before removal of GATC sites overlapping a CpG and GATC sites that are too close together (new Supplementary Fig. 3). Since the CpG density in CpG islands is high, filtering removes more GATC sites than in most other regions, but 21917 sites remain after filtering (55% of the total). After filtering, the GATC site density is 1.0 per kb, which is similar to promoters in general and about half that of gene bodies (2.0 per kb) (new Supplementary Fig. 3). (2) Centromeres have 45277 GATC sites before filtering and 31479 sites after filtering (70% remaining) (new Supplementary Fig. 3). If we hadn't eliminated the GATC sites overlapping with CpG from the calculation, disproportionately high CpG methylation would be expected to affect our results. We have added a brief comment to the paragraph at the top of page 5 to direct the reader to the GATC site data in new Supplementary Fig. 3.

4) There is no mention of telomeres within the study or discussion which I suspect is due to the lack of interpretable data from these regions. There would be added value to the discussion based on the author's insight into this genomic feature, building on their understanding of chromatin accessibility in living human cells, as informed by this study.

RESPONSE: The Reviewer is correct. There are only six GATC sites in the telomeric repeats, which we felt were not enough to give an overview of telomeric repeat accessibility.

(Remarks on code availability).

Reviewer #3 (Remarks to the Author)

I co-reviewed this manuscript with one of the reviewers who provided the listed reports. This is part of the Nature Communications initiative to facilitate training in peer review and to provide appropriate recognition for Early Career Researchers who co-review manuscripts.(Remarks on code availability)

I did not review the code.

Reviewer #4 (Remarks to the Author)

I co-reviewed this manuscript with one of the reviewers who provided the listed reports. This is part of the Nature Communications initiative to facilitate training in peer review and to provide appropriate recognition for Early Career Researchers who co-review manuscripts.(Remarks on code availability)

Reviewer #5 (Remarks to the Author):

How chromatin organization modulates DNA accessibility for gene regulation remains an important question in cell biology. Prajapati et al. utilized DNA adenine methyltransferase to address this question and investigated genome accessibility at all GATC sites in living MCF7 and MCF10A cells. The authors found the human genome was globally accessible in living cells and that heterochromatic GATC sites were methylated only more slowly than euchromatic sites. Interestingly, the centromeric sites were partly inaccessible. The authors' approach is unique, and their findings are exciting and valuable to a wide range of researchers. For publication in Nature Communications, several issues need to be addressed. My specific comments are as follows:

Major points:

(1) Figs. 1 and 2. The results are very impressive. However, I wonder how the frequencies (densities) of GATC sites differ among the analyzed regions, such as active gene regions (promoters and gene bodies), inactive gene regions (promoters and gene bodies), and other regions shown in panel h. Since the GATC frequency in each region may affect the results and their interpretation, a detailed comparison of these densities would be valuable. For example, the consensus sequence for the 171-bp alpha-satellite does not contain a GATC motif (PMID 2030938), suggesting that GATC sites are very rare in alpha-satellite regions.

RESPONSE: We have added a table listing the GATC site densities for various genomic regions (new Supplementary Fig. 3) and a brief comment on page 5 to direct the reader to these data. The GATC site densities (after filtering for sites that overlap CpG methylation sites and for sites that are too close together; see Methods) of active and inactive gene bodies are very similar (2.0 per kb for both) and higher than active (1.0 per kb) and inactive (1.6 per kb) promoters. Other regions: 2.5 per kb for tRNA genes, 2.0 per kb for gene bodies, 1.8 per kb for enhancers, 1.2 per kb for promoters, 1.0 per kb for CpG islands and 0.5 per kb for centromeres. Promoters and CpG islands are lower than gene

bodies mostly because they are enriched in CpG, increasing the probability of GATCG sites, which cannot be cut by DpnI if methylated (and so were excluded from our analysis). The centromeres are lower still because many of the 171-bp repeat elements lack a GATC site (as noted by the Reviewer). The sequence variation is such that the site density is 0.7 per kb (one GATC site per repeat would give a site density of ~5.8 per kb). After filtering, the site density decreases to 0.4 because some GATC sites are GATCG. This effect is particularly noticeable for the SF1 repeats (site density = 1.2 before filtering and 0.2 after filtering).

(2) Experimental conditions of live cells and isolated nuclei seem very different and are not comparable. To accurately compare methylation kinetics, the authors should quantify Dam expression levels in the cells.

RESPONSE: We agree that the experimental conditions for live cells and nuclei are not comparable. In response to Reviewer 1, who raised the same issue, we have inserted the following clarifying paragraph on page 8: "We note that the experimental conditions for Dam methylation in live cells and isolated nuclei should not be compared, because they are not the same type of experiment. In vivo, the Dam concentration increases very slowly with time after transduction (Fig. 1b; Supplementary Fig. 5b) and so does the amount of DNA substrate as the cells continue to divide, whereas in nuclei experiments, the Dam and nuclei concentrations are constant. This is an important distinction because the methylation rate should be directly proportional to the enzyme and substrate concentrations. Furthermore, there is wide variation in the time (in hours) before individual MCF7 cells become GFP-positive (as we observed in yeast cells (Prajapati et al. 2024))."

Using the only available commercial antibody against Dam (Thermo-Fisher MA1-12717), we could detect the Dam from NEB used in the nuclei experiments, but not Dam-3HA expressed in yeast cells, suggesting that the commercial Dam antibody is much less sensitive than the anti-HA antibody. However, for the reasons mentioned above, we don't think Dam quantification is necessary. The key points are the conclusions from each type of experiment: The live cell experiments show that Dam can access almost all the genome in vivo, whereas the nuclei experiments show that Dam access is strictly limited.

Furthermore, the nuclei isolation buffer contains spermidine, which can induce chromatin compaction. The results cannot exclude the possibility that the limited accessibility was due to the artificial compaction during the nuclei preparation.

RESPONSE: Our earlier experiments with yeast nuclei did not include spermidine in the buffer, with essentially the same result. However, we agree that this is an important point and so we have repeated the nuclei experiment in the absence of spermidine. We find the same result (now included in Supplementary Fig. 8); we have also added a scatter plot comparing the replicates to Supplementary Fig. 10.

(3) Line 491. The numbers of virus particles (MOI) used for cycling and confluent MCF10A cells were quite different. Could we compare the methylation kinetics in cycling/confluent MCF10A data? The authors need comparable expression levels of the Dam.

RESPONSE: We used twice as much virus for the confluent cells. We have added a comment in the Methods on page 10 (line 436): "(twice as much adenovirus was needed for confluent cells as for

log phase cells, possibly because confluent cells are smaller with less surface area exposed to the virus)." We could compare the methylation kinetics in cycling and confluent MCF10A cells. However, a serious complication is that the cycling cells are producing more substrate for Dam due to DNA replication.

(4) Fig. 4. I consider how Dam can access GATC sites in the genome very intriguing. Can nucleosomes block methylation of nucleosomal DNA except for linker DNA? It is known that H2A and H2B in the nucleosome are frequently replaced (PMID: 11425866). I wonder whether Dam could methylate nucleosomal DNA with only H3 and H4.

RESPONSE: Actually, this is precisely what we are proposing: nucleosomes block methylation in nuclei but not in vivo (Fig. 4e). In nuclei, only the linkers get methylated, resulting in a plateau, because nucleosomes are stable and nucleosomal DNA cannot be methylated. In vivo, all the DNA gets methylated, indicating that the nucleosomes are dynamic. We apologise for not making this clear; we hope that the new paragraph in the nuclei section on pages 7 and 8 will clarify our explanation of the difference between the results obtained for live cells and nuclei.

H2A/H2B replacement in vivo could be part of the explanation for nucleosome dynamics (the "on/off" model in Fig. 4e). We don't know if H3-H4 tetrasomes can be methylated by Dam.

(5) After extensive methylation of the genomic DNA, what impact did this have on chromatin structure and organization? Additionally, were the cells still viable following up to 72 hours of Dam expression?

RESPONSE: We have performed MNase-seq before and after 72 h of Dam expression. There is little impact on chromatin organisation. In response to Reviewer 1, we also did RNA-seq before and after virus treatment and observed little effect of Dam expression. The cells are ~90% viable after 72 h of treatment with the Dam-GFP virus or the GFP control virus, as shown by acridine orange/ propidium iodide staining. We have added new Supplementary Fig. 7 and inserted a paragraph at the top of page 7.

(6) Line 93. Unlike the yeast genome, which is fully methylated within several hours (Ref. 30), it took 48-72 hrs for human genomic DNA. Why? Is this because human genome chromatin is generally more condensed than yeast chromatin (PMID: 37385880)?

RESPONSE: Dam expression in MCF7 and MCF10A cells is relatively slow, taking at least 12 h after viral transduction before significant Dam is detected (Fig. 1b), whereas Dam methylation in yeast is much faster and complete in 4 h (Prajapati et al. 2024). For both cell types, there is a wide variation in the time it takes for individual cells to become GFP-positive. Also worth noting is the huge difference in genome size between diploid human cells (6 Gb) and haploid yeast (12.4 Mb), which means there are far more GATC sites to methylate in human cells (~14 million vs. ~35,000). As suggested by the Reviewer, the degree of condensation may also be a factor, but we can't compare rates in yeast and human cells because there are too many variables. We have added a comment on the slow rate of Dam expression in human cells and the cell-to-cell variation in expression timing at the end of the Results section on page 8.

Minor points:

(1) Lines 19-20 (also related to Line 329). "euchromatin, which is relatively decondensed... and heterochromatin, which is condensed..."

Recent studies have suggested that euchromatin can also be condensed, with only limited regions, such as enhancers and transcription start sites (TSSs) remaining decondensed (PMIDs: 32967822, 37018405, 37385880). It would be beneficial to include this updated perspective on chromatin structure.

RESPONSE: In this introductory statement, we are comparing euchromatin condensation to heterochromatin condensation at the level of nuclei in micrographs rather than at the level of the chromatin filament. We agree that euchromatin is at least partially condensed in nuclei and that its accessibility in nuclei is similar to that of heterochromatin. We now cite the interesting review from the Maeshima lab (PMID 37385880) and the Nozaki paper (PMID 37018405) to ensure that readers are up-to-date on this.

(2) Lines 68 and 305. To recruit ATP-dependent chromatin remodelers, local nucleosome dynamics in euchromatin and heterochromatin may be important. It would be valuable to discuss the nucleosome fluctuations in these regions (bioRxiv, 2024.10. 20.618801; PMID: 32574554).

RESPONSE: We think that local nucleosome dynamics may be part of the same phenomenon as global dynamics. We are very interested in trying to understand the role of local recruitment of ATP-dependent remodelers in global dynamics, but we would prefer not to speculate too much here.

RESPONSE TO REVIEWER 5

Reviewer #5 (Remarks to the Author):

The authors addressed most of my previous comments, and the revised manuscript is clearly improved. However, prior to publication in Nature Communications, there are a couple of remaining issues that still need to be addressed:

Comment 1:

While the authors presented the methylated fraction data, they should also provide data normalized by GATC site density, as this is essential for a fair comparison across genomic regions. This point was also raised by Reviewer #2.

RESPONSE: It is not appropriate to normalise our data to GATC site density. This is because we are considering individual sites, not the whole genomic region. In the case of the sequence-specific Dam methylase, only GATC sites are targets, not the whole region. We are measuring Dam methylation at each GATC site separately. For every GATC site, there is a set of 'fraction methylated' values for the time course, one for each time point. At each time point, there is one 'fraction methylated' value for every GATC site in the region; we calculate the median of these values. A region with a site density of 2 has twice as many GATC sites as a region with a density of 1, but it also has twice as many 'fraction methylated' values. We apologise for not making this clear in our original response to the Reviewer. We have added explanatory sentences on page 12, as follows:

"Dam methylation at each individual GATC site is measured separately. For every GATC site, there is a set of 'fraction methylated' values for the time course, one value for each time point. For analysis of GATC sites in a specific genomic region, we calculate the median of all 'fraction methylated' values for every GATC site in the region at each time point."

Comment 3:

To support the comparison of methylation kinetics between cycling and confluent MCF10A cells, the authors should directly compare Dam expression levels in these two conditions by western blot. Without confirming comparable expression levels, interpretation of kinetic differences remains uncertain.

RESPONSE: We compared the Dam expression levels for cycling and confluent cells in the same blot (both replicates), as requested (please see revised Supplementary Fig. 6b). We used the same samples that were shown in separate blots in original Supplementary Figures 5 and 6. There is some variation in the timing of expression, but the Dam expression levels are comparable in cycling and confluent cells.